# Missed opportunity for family planning counselling along the continuum of care in Arusha region, Tanzania

Caroline Amour[1]*, Rachel N. Manongi[2], Michael J. Mahande[1], Bilikisu Elewonibi[3], Amina Farah[4], Sia Emmanuel Msuya[1,2,4], Iqbal Shah[3]

1 Department of Epidemiology and Biostatistics, Institute of Public Health, Kilimanjaro Christian Medical University College, Moshi, Tanzania, 2 Department of Community Health, Institute of Public Health, Kilimanjaro Christian Medical University College, Moshi, Tanzania, 3 Department of Global Health and Population, Harvard University T.H. Chan School of Public Health, Boston, Massachusetts, United States of America, 4 Department of Community Health, KCMC Hospital, Moshi, Tanzania

* lyneamour@gmail.com

## Abstract

### Introduction

Adequate sexual and reproductive health information is vital to women of reproductive age (WRA) 15 to 49 years, for making informed choices on their reproductive health including family planning (FP). However, many women who interact with the health system continue to miss out this vital service. The study aimed to identify the extent of provision of FP counselling at service delivery points and associated behavioral factors among women of reproductive age in two districts of Arusha region. It also determined the association between receipt of FP counselling and contraceptive usage.

### Methods

Data were drawn from a cross-sectional survey of 5,208 WRA residing in two districts of Arusha region in Tanzania; conducted between January and May 2018. Multistage sampling technique was employed to select the WRA for the face-to-face interviews. FP counseling was defined as receipt of FP information by a woman during any visit at the health facility for antenatal care (ANC), or for post-natal care (PNC). Analyses on receipt of FP counseling were done on 3,116 WRA, aged 16–44 years who were in contact with health facilities in the past two years. A modified Poisson regression model was used to determine the Prevalence Ratio (PR) as a measure of association between receipt of any FP counseling and current use of modern contraception, controlling for potential confounders.

### Results

Among the women that visited the health facility for any health-related visit in the past two years, 1,256 (40%) reported that they received FP counselling. Among the women who had had births in the last 30 months; 1,389 and 1,409 women had contact with the service delivery points for ANC and PNC visits respectively. Of these 31% and 26% had a missed FP counseling at ANC and PNC visit respectively. Women who were not formally employed

**Data Availability Statement:** All relevant data are within the manuscript and S1 Data.

**Funding:** This project was made possible by a grant from an anonymous donor to Harvard T.H.

Chan School of Public Health. The funder had no role in study design, data collection and analysis, decision to publish, or preparation of the manuscript.

**Competing interests:** None of the research costs or authors' salaries were funded, in whole or in part, by a tobacco company. The identity of the donor is irrelevant to editors or reviewers' assessment of the validity of the work. The authors are not aware of any competing interests.

**Abbreviations:** cPR, Crude Prevalence ratio; aPR, Adjusted Prevalence ratio; CI, Confidence interval; CPR, Contraceptive prevalence rate; FP, Family planning; SSA, Sub-Saharan Africa; ANC, Antenatal Care; PNC, Post-natal care; PPFP, Post-partum Family Planning; WRA, Women of reproductive age; TDHS, Tanzania and demographic health survey; HF, Health facility; SDP, Service delivery point.

were more likely to receive FP counselling during facility visit than others. WRA who received any FP counseling at PNC were significantly more likely to report current use of modern contraception than those who did not (adjusted PR [adj. PR] = 1.28; 95% Confidence Interval [CI]: 1.09, 1.49).

## Conclusion

Overall, only 40% women reported that they received any form of FP counseling when they interfaced with the healthcare system in the past two years. Informally employed women were more likely to receive FP counselling, and women who received FP counselling during PNC visits were significantly more likely to use contraceptive in comparison to the women who did not receive FP counselling. This presents a missed opportunity for prevention of unintended pregnancies and suggests a need for further integration of FP counseling into the ANC and PNC visits.

## Introduction

Contraceptive counselling is a key intervention that could decrease unintended pregnancies by increasing contraceptive uptake and enabling effective contraceptive use. Contraceptive counselling has been shown to be associated with increased contraceptive use [1–4]. Several studies emphasize the importance of contraceptive counselling; it creates opportunities for a woman to obtain necessary information to make an informed decision that is appropriate for the reproductive needs and goals of the client [5–8]. Contraceptive counselling has also been shown to increase method continuation, reduce contraceptive switching/discontinuation and increase method satisfaction [9].

Compared with women in high income countries, women in low- and middle- income countries face greater risk of unwanted pregnancies and unsafe abortions which may lead to future health complications. One in three pregnancies in Tanzania are either unwanted or mistimed [10] contributing to an estimated 25,000 unsafe abortions annually. Many of these abortions occur in women under age 25 years [11]. Unsafe abortions contribute to 16% of maternal deaths [12]. Contraception is vital, given the high prevalence of unintended pregnancy and its impact on the health of the woman and child [13]. More than 30% of maternal and 10% of child mortality can be averted by the use of FP methods [14]. The most current estimate of modern contraceptive prevalence rate (mCPR) among married/ cohabiting women in Tanzania is 32% in 2015–2016 [10], well below the Tanzanian government's target of 45% target by 2020 [15].

Most women in Tanzania come in contact with the health care sector numerous times during their lifespan. According to the 2015 Tanzanian Demographic and Health Survey (TDHS), 98% of women who have had at least one pregnancy attended at least one antenatal care (ANC) visit, 64% deliver at a health facility, and the child vaccination coverage was greater than 95% in more than 90% of the districts [10]. This creates an opportunity for FP information to be disseminated during pre-natal, post-natal, and maternal health service delivery. Provision of FP information and counseling is an essential aspect of Tanzania maternal health plan and provided free of charge [16–18]. During these visits women are expected to receive information on FP methods that are available, possible side effects, and the importance of resuming the FP after delivery [1].

There is limited information on the prevalence of contraceptive counselling in Tanzania when women come into contact with health facilities (HFs). One study found that FP counselling is more likely to be offered in private service delivery points compared to public ones [19]. In response to the Tanzania FP 2020 commitment of reaching 45% modern contraceptive prevalence rate by 2020, this study determines the extent of provision of FP counselling at service delivery points, and its associated factors in Arusha. The study also assessed whether being counselled during clinic visit increases the use of modern contraceptives.

## Materials and methods

### Data collection

Data were drawn from a cross-sectional survey of WRA in two districts of Arusha region, northeastern Tanzania. The household survey data were collected from January to May 2018. The data were originally collected for an impact evaluation of a community-based intervention on contraceptive information, counselling and referral [20]. The survey was conducted on a representative sample of females aged 16–44 years, residing in the study area. A multi-stage random sampling was employed to select 3,938 women of reproductive age from Arusha city council and Meru district council. Further details on the study, participants and sampling strategy are described elsewhere [20].Women who consented to participate in the survey were asked questions relating to pregnancy history, use of FP services, and whether or not they had ever received FP counselling at a health facility. Only women who were sexually active and had visited a health facility in the last two years were included in the analysis (Fig 1).

### Study variables

**Outcome variable.** The outcome of interest in this analysis was whether or not a woman received FP counselling during any visit to a health facility (HF), during a visit for ANC and visit for PNC within the past two years. Women were included if they had any visit, and then a separate analysis was limited to those women who had had a birth in the last 30 months and who visited for ANC or PNC. Women were considered to have received FP counseling if, at any visit to the HF within the past two years, they reported that they received and/or discussed any information about FP with a health provider. The definition of counselling used was broad and did not specify the content of what the woman and her provider discussed. A missed opportunity for FP counseling was defined as a woman's ANC or PNC clinic visit at which a staff member did not provide any FP counseling.

In this analysis, current use of modern contraceptive was the secondary study outcome and was defined as currently using any of the following methods of FP to delay or limit pregnancy i.e. female sterilization, intra-uterine device, injectables, implants, oral contraceptive pills, condoms and emergency contraception. This variable was categorized as 1 if currently using any one of the modern contraceptive methods, and 0 if otherwise.

**Explanatory variables.** Explanatory variables in this analysis included socio-demographic characteristics and women characteristics, from all women enrolled in the above-mentioned survey.

Women's socio-demographic information (age as a continuous variable; marital status categorized as Never married (0),Married or cohabiting (1) and Divorced/widowed/separated (2); Occupation categorized as (0) for those employed or do any work and (1) otherwise; distance to health facility was categorized as less than 2 km (0), 2–3 km (1) and 4 km and above (2) and parity as a continuous variable were included in the analysis; with reference from other studies done on FP Counselling [14, 21]. To construct the wealth index, we identified variables in our

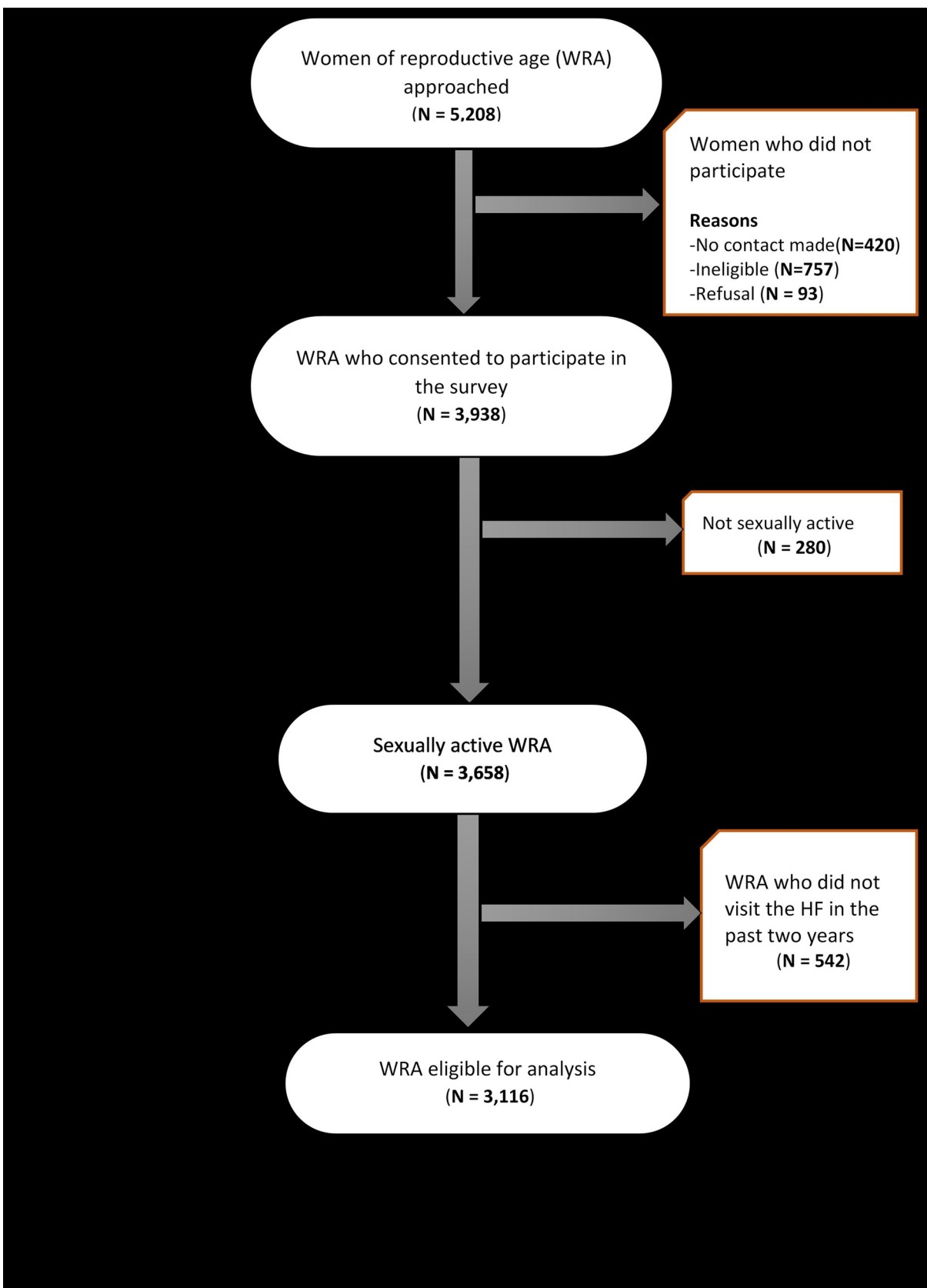

**Fig 1. Sample selection description flowchart.**

survey that are similar to the ones in the Tanzania Demographic and Health Survey (TDHS). It was created as a composite, asset-based measure from a series of questions and

then standardized to quintiles. Using these common variables, we categorized using the same cut-off points as the TDHS through principal components analysis. Hence, we ended up with five categories or groups of wealth index namely, poorest, poorer, medium, richer, and richest.

## Data analysis

Analyses were conducted on 3,116 women to provide descriptive statistics for women's socio-demographic characteristics and reproductive health characteristics. Bivariate regression analysis was used to examine predictors of FP counseling, if received during health facility visit. Multivariate regression analysis was used to determine the relationship between each predictor and the outcome of interest, controlling for the confounding effects of other factors. Modified Poisson regression was used to determine the association between receipt of FP counselling and current use of modern contraception, prevalence ratio (PR) was used as the measure of association. Modified Poisson regression directly estimates prevalence ratios and produces confidence intervals with the correct nominal coverage when individual-level data are available [22, 23]. We used sample weights calculated based on the multistage sampling design in all analyses. All data were analyzed using STATA SE 15.1 software (StataCorp, USA).

## Ethical consideration

Ethical approval was received from the National Institute of Medical Research (NIMR)—NIMRlHQIR.8cNol. 1/1171, the Kilimanjaro Christian Medical College Institutional Review Board– 2240 and Harvard T.H. Chan School of Public Health Institutional Review Board—RB17-1794 to conduct the household survey. A written participant consent was obtained before participation in the study for women aged 18 years and older. Participants who were aged 16–17 an assent was requested. Consent for them to participate was sought from partners for those who were married/ cohabiting, and from parents/ guardians for those who were under parental care. The women included in this study were only the ones who consented to participate in the study.

# Results

## Characteristics of the study participants

The sociodemographic, economic and health related characteristics of the study participants are shown in Table 1. A total of 3,116 women were included in the overall analysis. The mean (±standard deviation) age of the women was 30.3 (±6.9) years. Most of the women were between 20 and 29 years old (45%), had three or more children (37%) and were in the richer wealth quintile (46%). Three-quarter of the study participants were currently married or living with a partner (77%). Overall, among the women who had had births in the last 30 months, sixty nine percent received FP counselling at ANC and 74% at PNC visits.

## Predictors of receipt of family planning counseling along the continuum of care

A total of 462 women who were not pregnant in the past two years were excluded for the ANC and PNC analysis. The association between respondent's characteristics and FP counselling at the health facility in the past two years is shown in Table 2. Forty percent (1256) of women reported that they received FP counseling at any visit of the health facility in the past two

**Table 1. Characteristics of the study participants (N = 3,116).**

| Characteristics | n (%) |
|---|---:|
| Woman age (years) | |
| Below 20 | 109 (3.5) |
| 20–29 | 1396 (44.8) |
| 30–39 | 1239 (39.8) |
| 40 and above | 372 (11.9) |
| Marital status | |
| Never married | 417 (13.4) |
| Married/Cohabiting | 2389 (76.7) |
| Divorced/widowed/separated | 310 (9.9) |
| Work/Employed | |
| Yes | 1974 (63.3) |
| No | 1142 (36.7) |
| Wealth index | |
| Poorest | 518 (16.6) |
| Poorer | 271 (8.7) |
| Middle | 519 (16.7) |
| Richer | 1438 (46.1) |
| Richest | 370 (11.9) |
| Parity | |
| 0 | 462 (14.8) |
| 1 | 720 (23.1) |
| 2 | 780 (25.0) |
| 3 and above | 1154 (37.1) |
| Distance to nearest health facility (n = 1,109) | |
| <2 km | 583 (52.6) |
| 2_3 km | 255 (23.0) |
| 4 km and above | 271 (24.4) |
| Received FP counselling at any Health facility visit | |
| Yes | 1256 (40.3) |
| No | 1860 (59.7) |
| Received FP counselling at any ANC visit (n = 1,389) | |
| Yes | 956 (68.8) |
| No | 433 (31.2) |
| Received FP counselling at any PNC visit (n = 1,409) | |
| Yes | 1037 (73.6) |
| No | 372 (26.4) |

*% is the weighted percent

years. The receipt of FP counselling at any visit differed by employment status and wealth. A higher proportion of women who were not employed received FP counselling within the past 2 years of visiting the facilities compared to others (adjusted [adj.] PR: 1.24; 95% confidence interval [CI]: 1.20, 1.50).

**Association between receipt of any FP counseling and current use of modern contraception.** Table 3 shows the results of analysis of association between receipt of any FP counselling services and current use of modern contraceptive among women who visited a health facility in the past two years. After adjusting for woman's age, marital status and parity; the

**Table 2. Predictors for receiving FP counselling at any health facility (HF) visit (N = 3,116).**

| Characteristics | Counselled n (%) | Receipt of Family Planning counselling | | |
|---|---|---|---|---|
| | | Any HF Visit (n = 3,116) aPR (95% CI) | Any ANC visits (n = 1,389) aPR (95% CI) | Any PNC visits (n = 1,409) aPR (95% CI) |
| Woman age (years) | | | | |
| Below 20 | 27 (21.3) | Ref | Ref | Ref |
| 20–29 | 632 (45.3) | 0.95 (0.53–1.72) | 1.65 (0.57–4.76) | 1.22 (0.68–2.18) |
| 30–39 | 500 (38.4) | 0.70 (0.37–1.29) | 1.84 (0.63–5.32) | 1.25 (0.69–2.29) |
| 40 and above | 97 (23.9) | 0.61 (0.30–1.23) | 1.81 (0.61–5.34) | 1.53 (0.84–2.79) |
| Marital status | | | | |
| Single | 88 (20.1) | Ref | Ref | Ref |
| Married/Cohabiting | 1068 (43.7) | 1.27 (0.79–2.06) | 0.85 (0.71–1.03) | 0.84 (0.72–0.99) * |
| Formerly married | 100 (30.0) | 0.97 (0.54–1.76) | 0.88 (0.67–1.14) | 0.85 (0.67–1.07) |
| Work/Employed | | | | |
| Yes | 734 (35.3) | Ref | Ref | Ref |
| No | 522 (44.3) | 1.24 (1.02–1.50) * | 0.93 (0.82–1.06) | 0.93 (0.82–1.07) |
| Wealth index | | | | |
| Poorest | 229 (43.6) | 1.19 (0.91–1.56) | 1.16 (0.99–1.35) | 0.99 (0.82–1.19) |
| Poorer | 128 (41.9) | 1.36 (1.02–1.80) * | 1.16 (0.96–1.41) | 1.14 (0.96–1.34) |
| Middle | 217 (40.8) | 1.11 (0.89–1.43) | 1.08 (0.91–1.28) | 1.05 (0.89–1.23) |
| Richer | 554 (36.1) | Ref | Ref | Ref |
| Richest | 128 (37.3) | 1.19 (0.89–1.61) | 1.17 (1.00–1.36) * | 1.11 (0.95–1.29) |
| Parity | | | | |
| 1 | 339 (47.3) | Ref | Ref | Ref |
| 2 | 342 (41.1) | 0.95 (0.74–1.21) | 0.91 (0.76–1.12) | 1.04 (0.90–1.21) |
| 3 and above | 500 (41.0) | 1.15 (0.88–1.50) | 0.97 (0.80–1.17) | 0.98 (0.81–1.20) |
| Distance to nearest health facility | | | | |
| <2 km | 232 (41.2) | Ref | Ref | Ref |
| 2_3 km | 135 (49.9) | 1.19 (0.96–1.47) | 0.97 (0.84–1.12) | 1.07 (0.94–1.22) |
| 4 km and above | 130 (43.5) | 1.02 (0.81–1.28) | 0.93 (0.80–1.09) | 1.04 (0.92–1.18) |

Statistically significant at

*P<0.05

**P< 0.001

women who received any FP counselling services at a health facility visit, had a 4% higher current usage of modern contraceptive in comparison to those who did not, although this association was not statistically significant (adjusted [adj.] PR: 1.04; 95% confidence interval [CI]: 0.92, 1.16). Current use of a modern contraceptive method was 31% more prevalent among women who received FP counselling at PNC visit compared to those who didn't, this association was statistically significant (PR: 1.31; 95% confidence interval [CI]: 1.11, 1.54).

## Discussion

Our study of missed opportunities for FP counseling among WRA along the continuum of care highlights a few important findings. About 61% of the women in our sample who had an interaction with the healthcare system in the past 2 years did not receive any form of contraceptive counselling. Women not formally employed and those in the poorer wealth quintile

**Table 3. Receipt of FP counselling associated with current use of modern contraception (n = 3,116).**

|  | cPR (95% CI) | P-value | aPR (95% CI) | P-value |
|---|---|---|---|---|
| Received FP counselling at any Health facility visit |  |  |  |  |
| Yes | 1.48 (1.35–1.63) | <0.0001 | 1.04 (0.92–1.16) | 0.554 |
| No | Ref |  | Ref |  |
| Received FP counselling at any ANC visit |  |  |  |  |
| Yes | 1.29 (1.11–1.49) | 0.001 | 1.12 (0.97–1.28) | 0.124 |
| No | Ref |  | Ref |  |
| Received FP counselling at any PNC visit |  |  |  |  |
| Yes | 1.45 (1.23–1.73) | <0.0001 | 1.31 (1.11–1.54) | 0.001 |
| No | Ref |  | Ref |  |

*adjusted by woman's age, marital status and parity

predicted receiving FP counseling during any health facility visit. Counseling during PNC period increased the prevalence of modern contraceptive use.

We found that the receipt of FP counseling was 40% among the women who attended the health facility for any health visit. Among the women who visited the health facility for any visit and had a birth in the last 30 months, 68.8% received FP counselling during ANC and 73.6% at postnatal care visit. Availability of focused ANC and PNC care guidelines that stipulates what should be done at each visit including FP counseling may explain higher counseling reports than general Outpatient department (OPD)/ other clinic visits. These findings imply that 60% of the women missed FP counselling when they visited the facility for any health reason; while 31.2% and 26.4% missed FP counselling at ANC and post-natal care (PNC) visits respectively. Similarly findings have been reported elsewhere [8, 14, 16, 18, 24]. Kanyangara et al reported a 6% missed opportunity of FP counselling in sub-Saharan country sites. This difference from our study could be because in the other studies provision of FP counselling was reported by the facility-in-charge [25]. Overall, our findings highlight a missed opportunity for FP counselling among women, when coming into contact with a health facility. Hence, we suggest a vital need for a full-scale integration of FP counselling in any visit at the health facility along the continuum of care, especially at the PNC visits and child health services such as child immunization visit.

In Tanzania, 98% of women attend ANC care services provided by skilled provider [26]. ANC visits provides an opportunity for health worker to advice and counsel on the importance of FP. However, our study found that 31% of women were not counseled on FP at ANC and 26% at PNC visit. Those who were counseled during postnatal care visit were more likely to currently use modern contraceptive than those who were not counseled. This finding is similar to a study done in Ethiopia which found that FP planning counseling increased the chance of using postpartum family planning (PPFP) [27]. This is also similar to a study done in Kenya, where they found that contraceptive usage was 35% higher among women who had FP counselling during PNC [28, 29]. Lack of association between FP counseling during ANC visits and current usage of modern contraceptive has also been observed in other East African studies [16, 19]. It maybe that after successful delivery, women are more receptive to the information that will improve their own health and that of the new baby, hence higher prevalence of contraceptive use after being counselled during postnatal period [16, 19, 30]. Improved uptake of maternal, newborn and child health interventions after counseling during postnatal period have also been reported in breastfeeding and vaccination [31].

We also found that married or cohabiting women were 16% less likely to receive contraceptive counselling at PNC visits compared to women who were not married. This could be explained by the providers' perceptions of women's need for FP methods. A study in Kenya looked on provider-imposed barriers to FP use and found that providers reported to impose FP method restrictions on older women [32]. This association remained statistically significant after adjusting for other socio-demographics. These findings are contrary to a study in Ethiopia and Philippines, were receipt of contraceptive FP was positively associated with the women's marital status; where married women were more likely to receive information or be counselled on contraceptive usage [14, 21]. Studies in Tanzania and elsewhere have shown married/ cohabiting women are more likely to use contraceptives than single, divorced or separated women[14, 28, 33]. The difference in these studies to our study may be explained by the providers' perspective on married women usage of contraceptives. It may be that providers focus their attention to those who are at higher risk of not using the methods, thinking the other group will maintain their behavior.

## Strength and limitations

Our study addressed one of the prominent areas in numerous efforts to achieve the FP 2020 commitment aiding in reducing the unmet need for FP among WRA. This study had some limitations. Our study cannot infer a causal relationship between the association of FP counselling and current usage of modern contraceptive. Also, some women might have some recall bias about the content of what was discussed during ANC or PNC visits and thus under-report contraceptive counseling. In addition, the results cannot be generalized to the general population in Tanzania. This is because WRA who interacted with the health care system, attended ANC and PNC services could be different from those who do not utilize these services. Another limitation is that counseling was reported by women, no observations were done. Further, health facility factors like providers knowledge and skills in counseling and FP as well as availability of supplies was not assessed. We also cannot fully explain why some women did not use contraception even after receiving FP counselling.

However, our study suggests that FP counselling during PNC visit is an important determinant of current contraceptive usage among WRA as the receipt of FP counselling during PNC visit was significantly associated with higher prevalence of current contraceptive usage.

## Conclusion

Forty-percent of the women who had interacted with the service delivery points at any health visit had missed the opportunity for contraceptive counselling. Women with potential unmet need for contraceptive usage, had missed the opportunity for important information and counselling to prevent unwanted and mistimed pregnancies. It becomes even more prominent that women who received FP counselling during PNC visits were significantly more likely to use contraceptive in comparison to the women who did not receive FP counselling. Hence efforts should be directed at counselling women about the importance of contraception to avoid unwanted pregnancies. We emphasis particular recommendation on the need to enhance integration of FP into the continuum of care, in order to increase access to FP information and education, counseling and ultimately contraceptives use among WRA.

## Supporting information

**S1 Data.**
(DTA)

## Acknowledgments

The authors thank the Arusha Regional Medical Officer, District Medical Officers and other administrative staff in Arusha city and Meru district; for their cooperation and support during the study period. We also thank the study participants for their valuable participation in this study. Furthermore, we would like to thank the research assistants for their positive participation and involvement in this study.

## Author Contributions

**Conceptualization:** Michael J. Mahande, Bilikisu Elewonibi, Sia Emmanuel Msuya, Iqbal Shah.

**Data curation:** Caroline Amour, Michael J. Mahande, Sia Emmanuel Msuya.

**Formal analysis:** Caroline Amour.

**Funding acquisition:** Sia Emmanuel Msuya, Iqbal Shah.

**Investigation:** Bilikisu Elewonibi, Sia Emmanuel Msuya, Iqbal Shah.

**Methodology:** Caroline Amour, Michael J. Mahande, Bilikisu Elewonibi, Sia Emmanuel Msuya.

**Project administration:** Rachel N. Manongi, Bilikisu Elewonibi, Amina Farah, Iqbal Shah.

**Supervision:** Caroline Amour, Rachel N. Manongi.

**Validation:** Michael J. Mahande.

**Visualization:** Caroline Amour.

**Writing – original draft:** Caroline Amour.

**Writing – review & editing:** Caroline Amour, Rachel N. Manongi, Michael J. Mahande, Bilikisu Elewonibi, Amina Farah, Sia Emmanuel Msuya, Iqbal Shah.

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
