## [Decision Letter · Decision Letter 0]

15 Jan 2021

PONE-D-20-36870

Missed opportunity for family planning counselling along the continuum of care in Arusha region, Tanzania

PLOS ONE

Dear Dr. %Caroline Amour%,

Thank you for submitting your manuscript to PLOS ONE. After careful consideration, we feel that it has merit but does not fully meet PLOS ONE’s publication criteria as it currently stands. Therefore, we invite you to submit a revised version of the manuscript that addresses the points raised during the review process.

Please address each comment made by both reveiwers.

We look forward to receiving your revised manuscript.

Kind regards,

Mary Hamer Hodges, MBBS MRCP DSc

Academic Editor

PLOS ONE

Journal Requirements:

2. Thank you for stating in your financial disclosure: 

"This project was made possible by a grant from an anonymous donor to Harvard T.H. Chan School of Public Health. The funder had no role in study design, data collection and analysis, decision to publish, or preparation of the manuscript."

PLOS ONE requires you to include in your manuscript further information about the funder so that any relevant competing interests can be assessed. Please respond to the following questions:

Please state whether any of the research costs or authors' salaries were funded, in whole or in part, by a tobacco company (our policy on tobacco funding is at http://journals.plos.org/plosone/s/disclosure-of-funding-sources) Please state whether the donor has any competing interests in relation to this work (see http://journals.plos.org/plosone/s/competing-interests) .Please state whether the identity of the donor might be considered relevant to editors or reviewers’ assessment of the validity of the work.If the donors have no perceived or actual competing interests, please state: “The authors are not aware of any competing interests”.

This information should be included in your cover letter. We will amend your financial disclosure and competing interests on your behalf.

4. Please consider whether participants younger than 18 years old should be referred to as 'girls' rather than 'women'.

5. We noticed you have some minor occurrence of overlapping text with the following previous publication(s), which needs to be addressed:

- Nabirye, J., Matovu, J.K.B., Bwanika, J.B. et al. Missed opportunities for family planning counselling among HIV-positive women receiving HIV Care in Uganda. BMC Women's Health 20, 91 (2020). https://doi.org/10.1186/s12905-020-00942-6

In your revision ensure you cite all your sources (including your own works), and quote or rephrase any duplicated text outside the methods section. Further consideration is dependent on these concerns being addressed.

Reviewers' comments:

Reviewer's Responses to Questions

**Comments to the Author**

1. Is the manuscript technically sound, and do the data support the conclusions?

Reviewer #1: Partly

Reviewer #2: Yes

2. Has the statistical analysis been performed appropriately and rigorously? 

Reviewer #1: Yes

Reviewer #2: Yes

3. Have the authors made all data underlying the findings in their manuscript fully available?

Reviewer #1: Yes

Reviewer #2: No

4. Is the manuscript presented in an intelligible fashion and written in standard English?

Reviewer #1: Yes

Reviewer #2: Yes

5. Review Comments to the Author

Reviewer #1: Missed opportunity for family planning counselling along the continuum of care in Arusha region, Tanzania.

Abstract

L 6&7: Associated factors of what: behavioural of the women or the health provider? What about health worker knowledge and skills, commodity supply, distance to the health facility, please clarify

Please define women of reproductive age (15-49 years of age).

Methods

L 11: I suggest ‘face-to-face’

L 11: Continue with to the abbreviation ‘FP’ after it has been introduced

L 13: The FP counselling: is it a group or one-on-one (confidential) counselling during the clinic visit?

L 16: Would contraceptive prevalence rate (CPR) or even modern CPR (mCPR) be more appropriate?

Results

L 18: The percentage is sometime cited as ‘38%’ and sometime as "40%" Please clarify

Conclusion

L 29: What about the other associated factors that is mentioned in the introduction. Were there any outcomes?

Introduction

L 12-19: I suggest taking above before the background reference status of FP situation in TZ

Page 4, L 13: the study also assesse’d’ whether…

Materials and methods

L 16: Please stay with the abbreviation WRA

L 19: reference the data source: e.g. ministry of health or other partners

L 21: I suggest to delete "the"

Page 5, L 15: Are these referring to the "other associated factors"?

L 16, L17 and L19: keep to the abbreviation introduced "FP"

L 21: the outcome of L18-21 did not also come out in the conclusion

L 22: I suggest to include ‘as shown in fig 1’.

Sample selection description flow chart

L 1: FP counselling of (1,256/3,116) is 40% not 38%.

L7-10, please use a space before the opening bracket

If the analysis is done up this category, the coverage should be 40% instead of 38%.

Characteristic of the study participants

L 7: This is confusing, if 462 was excluded then, does it mean analysis for coverage was done on "2,654" out of the 3,116, then coverage of 38% is still questionable.

Table 1: Characteristic of the study participants (N=3,116)

This should be discussed as one of the associated factors to FP counselling coverage in the conclusion

Please double check your subtotals, for example age=3115, marital status=3,115, work/employed only = 3114, etc

Predicator of receipt of family planning counselling along the continuum of care

L 2: keep to the abbreviation "FP"

L 3: it is correctly stated here but in the Abstract it is stated as 38%. Please don’t use the word ‘about’ so the sentence needs reframing.

Discussion

L 4: Staying with the abbreviation WRA

L 7: Please discuss further, as table 1 show combined % for poorest and poorer as 24.2%, does this mean low FP counselling coverage is found among the middle to richest category; this also goes for employment.

L 10: here again the 38% instead of 40%

Strength and Limitations

L 5: If these factors were not assessed, then why are they mentioned in the abstract "aim" to identify the associated factors.

Conclusion

L 11: Here again the coverage is not consistent in the other section such as the conclusion in the abstract.

Reviewer #2: The title refers to continuum of care while only ANC and postpartum periods are considered! It is not clear how 38% was calculated since 3115 women were included in the analysis "Overall, 1,256 (38%) women reported that they received FP counseling at any of the health facility visits in the past two years. Of the 1,389 and 1,409 women who had contact with the service delivery points for ANC and PNC visits respectively; 27% and 26% had a missed FP counseling at ANC and PNC visit

respectively." Calculations of 27 and 26% are also not clear! "The data were originally collected for an impact evaluation of a community-based intervention on contraceptive information, counselling and referral." Are the results presented in this paper before the intervention or after the intervention? "A total of 3,116 women were included in the overall analysis; whereas 462

8 women who had no pregnancy were excluded for the ANC and PNC analysis." In figure 1, WRA included in the analysis was 3115, where did the extra participant come from to make them 3116? Where did the 462 women excluded come from since in figure 1, they are not shown and pregnancy was not mentioned as an exclusion criteria? "Overall, among the women who had

had births in the last 30 months more than seventy percent received FP counselling at ANC and PNC visits (73% and 74% respectively)." The reported percentage in the results section are different from those in the abstract, since it is reported that only 38% received FP counselling. In the discussion, it is not clear what the authors are saying in this statement "These findings imply that 61% of the women missed FP counselling when they visited the facility for any health reason; while 26.8% and 26.2% missed FP counselling at 1 ANC and post-natal care (PNC) visits respectively." The discussion needs to be redone once the results have been revised.

6. PLOS authors have the option to publish the peer review history of their article (what does this mean?). If published, this will include your full peer review and any attached files.

Reviewer #1: No

Reviewer #2: No

---

## [Author Response · Author response to Decision Letter 0]

19 Mar 2021

Point by point response to reviewer’s comments

Manuscript ID: Ms. [PONE-D-20-36870] 

Missed opportunity for family planning counselling along the continuum of care in Arusha region, Tanzania

Journal: PLOS ONE

Subject: Author's response to reviewer’s comments

Dear Editor, 

Thanks very much for the invitation to resubmit our manuscript entitled: “Missed opportunity for family planning counselling along the continuum of care in Arusha region, Tanzania”. We appreciate the reviewers’ useful comments and suggestions to improve our manuscript. We were able to address all the comments and suggestions, and therefore we are expecting that the paper can now be accepted in your journal. Please find our responses as are indicated in a red color. All comments requested have been incorporated in the main text.

Regarding Financial disclosures; None of the research costs or authors' salaries were funded, in whole or in part, by a tobacco company. The identity of the donor is irrelevant to editors or reviewers’ assessment of the validity of the work. The authors are not aware of any competing interests.

Caroline Amour

Corresponding author

Journal Requirements:

Thanks for this important observation. We have now made the necessary changes in the revised manuscript.

2. Thank you for stating in your financial disclosure: 

"This project was made possible by a grant from an anonymous donor to Harvard T.H. Chan School of Public Health. The funder had no role in study design, data collection and analysis, decision to publish, or preparation of the manuscript."

PLOS ONE requires you to include in your manuscript further information about the funder so that any relevant competing interests can be assessed. Please respond to the following questions:

1. Please state whether any of the research costs or authors' salaries were funded, in whole or in part, by a tobacco company (our policy on tobacco funding is at http://journals.plos.org/plosone/s/disclosure-of-funding-sources) 

2. Please state whether the donor has any competing interests in relation to this work (see http://journals.plos.org/plosone/s/competing-interests) .

3. Please state whether the identity of the donor might be considered relevant to editors or reviewers’ assessment of the validity of the work.

4. If the donors have no perceived or actual competing interests, please state: “The authors are not aware of any competing interests”.

This information should be included in your cover letter. We will amend your financial disclosure and competing interests on your behalf.

Thank you for the comment. None of the research costs or authors' salaries were funded, in whole or in part, by a tobacco company. The identity of the donor is irrelevant to editors or reviewers’ assessment of the validity of the work. The authors are not aware of any competing interests. This additional information has been updated in the cover letter and the revised manuscript.

Thank you for this important observation. A written participant consent was obtained before participation in the study for women aged 18 years and older. Participants who were aged 16 – 17 an assent was requested. Consent for them to participate was sought from partners for those who were married/ cohabiting, and from parents/ guardians for those who were under parental care. This information has been added to the methods section of the revised manuscript.

4. Please consider whether participants younger than 18 years old should be referred to as 'girls' rather than 'women'.

 We thank the editor for this important observation. We have now addressed this in the revised manuscript.

5. We noticed you have some minor occurrence of overlapping text with the following previous publication(s), which needs to be addressed:

- Nabirye, J., Matovu, J.K.B., Bwanika, J.B. et al. Missed opportunities for family planning counselling among HIV-positive women receiving HIV Care in Uganda. BMC Women's Health 20, 91 (2020). https://doi.org/10.1186/s12905-020-00942-6

In your revision ensure you cite all your sources (including your own works), and quote or rephrase any duplicated text outside the methods section. Further consideration is dependent on these concerns being addressed.

 The authors thank the editor for this observation. We have now made the necessary citations in the revised manuscript.

 

Comments to the Author

1. Is the manuscript technically sound, and do the data support the conclusions?

Reviewer #1: Partly

Reviewer #2: Yes

 2. Has the statistical analysis been performed appropriately and rigorously? 

 Reviewer #1: Yes

Reviewer #2: Yes

 3. Have the authors made all data underlying the findings in their manuscript fully available?

 Reviewer #1: Yes

Reviewer #2: No

 4. Is the manuscript presented in an intelligible fashion and written in standard English?

 Reviewer #1: Yes

Reviewer #2: Yes

 5. Review Comments to the Author

 Reviewer #1: Missed opportunity for family planning counselling along the continuum of care in Arusha region, Tanzania.

Abstract

L 6&7: Associated factors of what: behavioral of the women or the health provider? What about health worker knowledge and skills, commodity supply, distance to the health facility, please clarify

Please define women of reproductive age (15-49 years of age).

We thank the reviewer for the observation, we have made the additions and defined the women of reproductive age. The authors acknowledge that apart from the women’s behavioral factors studied in this paper, there could be other factors that could contribute. However, the Health worker and/or health facility related factors were not observed or studied at this time in this study project. 

Methods

L 11: I suggest ‘face-to-face’

We thank the reviewer for the comment, we have made the revision

L 11: Continue with to the abbreviation ‘FP’ after it has been introduced

We thank the reviewer for the comment, we have made the revision

L 13: The FP counselling: is it a group or one-on-one (confidential) counselling during the clinic visit?

We thank the reviewer for this important comment. FP counselling is done on a one-on-one basis between health care worker and client. Although in some instances general information on FP can be provided as group, but later on detailed information and client is given time to ask any questions or concerns.

L 16: Would contraceptive prevalence rate (CPR) or even modern CPR (mCPR) be more appropriate?

We thank the reviewer for this comment. Contraceptive Prevalence Rate (CPR) or modern CPR (mCPR) will tell us the extent or current usage of contraceptive or modern contraceptive. To determine the association between contraceptive usage and receipt of FP counselling, we implored a modified poisson regression model, whereas Prevalence Ratio (PR) was estimated.

Results

L 18: The percentage is sometime cited as ‘38%’ and sometime as "40%" Please clarify

We thank the reviewer for the comment, we have made the revision

Conclusion

L 29: What about the other associated factors that is mentioned in the introduction. Were there any outcomes?

We thank the reviewer for the comment, we have made the revision

Introduction

L 12-19: I suggest taking above before the background reference status of FP situation in TZ

We thank the reviewer for the comment, we have made the changes

Page 4, L 13: the study also assesse’d’ whether…

We thank the reviewer for the comment, we have made the revision

Materials and methods

L 16: Please stay with the abbreviation WRA

We thank the reviewer for the comment, we have made the editions in the revised manuscript

L 19: reference the data source: e.g. ministry of health or other partners

We thank the reviewer for the comment, we have added the reference

L 21: I suggest to delete "the"

We have made the deletion, thanks for the observation

Page 5, L 15: Are these referring to the "other associated factors"?

Thanks for the comment. Other associated factors include the demographics and other behavioral characteristics of the women.

L 16, L17 and L19: keep to the abbreviation introduced "FP"

We thank the reviewer for the comment, we have made the revisions

L 21: the outcome of L18-21 did not also come out in the conclusion

Thanks for the comment. The secondary outcome contraceptive usage was used to determine the association between receipt of FP counselling and current contraceptive usage. We found a significant association between receipt of FP counselling at any PNC visit and current usage of contraception (see Table 3). Conclusion was revised.

L 22: I suggest to include ‘as shown in fig 1’.

We thank the reviewer for the important observation. We have placed the figure in the data collection section, where sample selection was described.

Sample selection description flow chart

L 1: FP counselling of (1,256/3,116) is 40% not 38%.

We thank the reviewer for the comment, we have made the revisions

L7-10, please use a space before the opening bracket

If the analysis is done up this category, the coverage should be 40% instead of 38%.

We thank the reviewer for the comment, we have made the revisions

Characteristic of the study participants

L 7: This is confusing, if 462 was excluded then, does it mean analysis for coverage was done on "2,654" out of the 3,116, then coverage of 38% is still questionable.

We thank the reviewer for the comment. The 462 women who were never pregnant in the past two years, were excluded in the separate analysis that looked on predictors of FP counselling at any ANC and PNC visits. The revisions are made in the revised manuscript.

Table 1: Characteristic of the study participants (N=3,116)

This should be discussed as one of the associated factors to FP counselling coverage in the conclusion

We thank the reviewer for the comment, we have made the revisions

Please double check your subtotals, for example age=3115, marital status=3,115, work/employed only = 3114, etc

Thanks for the comment, we have double checked the subtotals. Age and marital status had one missing value in these variables, where employment status had two missing values.

Predictor of receipt of family planning counselling along the continuum of care

L 2: keep to the abbreviation "FP"

We thank the reviewer for the comment, we have made the revision

L 3: it is correctly stated here but in the Abstract, it is stated as 38%. Please don’t use the word ‘about’ so the sentence needs reframing.

We thank the reviewer for the comment, we have made the revision

Discussion

L 4: Staying with the abbreviation WRA

We have made the revision, thanks for the comment.

L 7: Please discuss further, as table 1 show combined % for poorest and poorer as 24.2%, does this mean low FP counselling coverage is found among the middle to richest category; this also goes for employment.

We thank the reviewer for the comment, we have revised the discussion to include detailed information in the revised manuscript.

L 10: here again the 38% instead of 40%

We thank the reviewer for the comment, we have made the revision

Strength and Limitations

L 5: If these factors were not assessed, then why are they mentioned in the abstract "aim" to identify the associated factors.

Thanks for the comment. The study assessed the predictors of receiving FP counselling and determined to see if there is any association between receiving of FP counselling and Current contraceptive usage. However, the study did not look at the reasons why some women did not use contraception even after receiving FP counselling.

Conclusion

L 11: Here again the coverage is not consistent in the other section such as the conclusion in the abstract.

We thank the reviewer for the comment, we have made the revision

Reviewer #2: 

The title refers to continuum of care while only ANC and postpartum periods are considered! Thanks for the comment. The study assessed FP counselling at any health facility visit, at any ANC visit and at any PNC visit in the past two years.

It is not clear how 38% was calculated since 3115 women were included in the analysis "Overall, 1,256 (38%) women reported that they received FP counseling at any of the health facility visits in the past two years. 

We thank the reviewer for the comment, we have made the revision in the revised manuscript.

Of the 1,389 and 1,409 women who had contact with the service delivery points for ANC and PNC visits respectively; 27% and 26% had a missed FP counseling at ANC and PNC visit respectively." Calculations of 27 and 26% are also not clear!

We thank the reviewer for the comment, we have made the revision in the revised manuscript after double checking the calculations.

"The data were originally collected for an impact evaluation of a community-based intervention on contraceptive information, counselling and referral." Are the results presented in this paper before the intervention or after the intervention?

Thanks for the question. The results presented in the paper were for the baseline data before the intervention.

"A total of 3,116 women were included in the overall analysis; whereas 462 women who had no pregnancy were excluded for the ANC and PNC analysis." In figure 1, WRA included in the analysis was 3115, where did the extra participant come from to make them 3116? 

Thanks for the comment. Figure one shows a total of 3,116 WRA included in the analysis (see Fig 1)

Where did the 462 women excluded come from since in figure 1, they are not shown and pregnancy was not mentioned as an exclusion criteria? 

We thank the reviewer for the comment. The 462 women who were never pregnant in the past two years, were excluded in a separate sub-analysis that looked on predictors of FP counselling at any ANC and PNC visits. The clarifications are made in the revised manuscript.

"Overall, among the women who had had births in the last 30 months more than seventy percent received FP counselling at ANC and PNC visits (73% and 74% respectively)." The reported percentage in the results section are different from those in the abstract, since it is reported that only 38% received FP counselling. 

We thank the reviewer for this important observation. Among the women that visited the health facility for any health-related visit in the past two years, 40% received FP counselling. Among the women who had had births in the last 30 months; 1,389 and 1,409 women had contact with the service delivery points for ANC and PNC visits respectively. Of these 68.8% and 73.6% received FP counselling at ANC and PNC visits respectively. Necessary revisions were made in the revised manuscript.

In the discussion, it is not clear what the authors are saying in this statement "These findings imply that 61% of the women missed FP counselling when they visited the facility for any health reason; while 26.8% and 26.2% missed FP counselling at 1 ANC and post-natal care (PNC) visits respectively." The discussion needs to be redone once the results have been revised.

We thank the reviewer for this important observation. Among the women that visited the health facility for any health-related visit in the past two years, only 40% received FP counselling. This means that 60% of the women that visited the health facility for any health-related visit did not get FP counselling. Among the women who had had births in the last 30 months; 1,389 and 1,409 women had contact with the service delivery points for ANC and PNC visits respectively. Of these 31.2% and 26.4% missed FP counselling at ANC and PNC visits respectively. Necessary revisions were made in the revised manuscript.________________________________________

---

## [Decision Letter · Decision Letter 1]

19 Apr 2021

Missed opportunity for family planning counselling along the continuum of care in Arusha region, Tanzania

PONE-D-20-36870R1

Dear Dr. %Caroline Amour%,

We’re pleased to inform you that your manuscript has been judged scientifically suitable for publication and will be formally accepted for publication once it meets all outstanding technical requirements.

Kind regards,

Mary Hamer Hodges, MBBS MRCP DSc

Academic Editor

PLOS ONE

Additional Editor Comments (optional):

Reviewers' comments:

Reviewer's Responses to Questions

**Comments to the Author**

1. If the authors have adequately addressed your comments raised in a previous round of review and you feel that this manuscript is now acceptable for publication, you may indicate that here to bypass the “Comments to the Author” section, enter your conflict of interest statement in the “Confidential to Editor” section, and submit your "Accept" recommendation.

Reviewer #1: All comments have been addressed

Reviewer #2: All comments have been addressed

2. Is the manuscript technically sound, and do the data support the conclusions?

Reviewer #1: Yes

Reviewer #2: Yes

3. Has the statistical analysis been performed appropriately and rigorously? 

Reviewer #1: Yes

Reviewer #2: Yes

4. Have the authors made all data underlying the findings in their manuscript fully available?

Reviewer #1: Yes

Reviewer #2: Yes

5. Is the manuscript presented in an intelligible fashion and written in standard English?

Reviewer #1: Yes

Reviewer #2: Yes

6. Review Comments to the Author

Reviewer #1: Page 1, Corresponding Author: "UNITED REPUBLIC OF" a word or two is missing here

Discussion

L12: keep to "FP" be consistent keep to the abbreviation with the document

Result

The comments are few, see attached file, particularly table 1

Reviewer #2: All the comments have been addressed adequately. I note that the numbers in the paper are now harmonised. This has improved the flow and understandability of the paper.

7. PLOS authors have the option to publish the peer review history of their article (what does this mean?). If published, this will include your full peer review and any attached files.

Reviewer #1: No

Reviewer #2: No

---

## [Editor Report · Acceptance letter]

2 Jul 2021

PONE-D-20-36870R1 

Missed opportunity for family planning counselling along the continuum of care in Arusha region, Tanzania 

Dear Dr. Amour:

I'm pleased to inform you that your manuscript has been deemed suitable for publication in PLOS ONE. Congratulations! Your manuscript is now with our production department. 

Kind regards, 

on behalf of

Dr. Mary Hamer Hodges 

Academic Editor

PLOS ONE